# Different race pacing strategies among runners covering the 2017 Berlin Marathon under 3 hours and 30 minutes

Iker Muñoz-Pérez[1,2], Marcos Mecías-Calvo[1,3]*, Jorge Crespo-Álvarez[1,4], María Luisa Sámano-Celorio[5], Pablo Agudo-Toyos[1], Carlos Lago-Fuentes[1]

1 Facultad de Ciencias de la Salud, Universidad Europea del Atlántico (UNEATLANTICO), Santander, Spain, 2 Runnea, Barakaldo, Spain, 3 Centro de Investigación y Tecnología Industrial de Cantabria (CITICAN), Santander, Spain, 4 Recursos de Obras, Montajes y Asistencias (ROMA), Santa Cruz de Bezana, Spain, 5 Universidad Internacional Iberoamericana, Campeche, México

* marcos.mecias@uneatlantico.es

**Data Availability Statement:** All data files are available from the Berlin Marathon database web:

## Abstract

The purposes of this study were 1) to analyse the different pacing behaviours based on athlete's performance and 2) to determine whether significant differences in each race split and the runner's performance implied different race profiles. A total of 2295 runners, which took part in Berlin's marathon (2017), met the inclusion criteria. 4 different groups were created based on sex and performance. Men: Elite (<02:19:00 h), Top 1 (<02:30:00 h), Top 2 (<02:45:00 h) and Top 3 (<03:00:00 h); women: Elite (02:45:00 h), Top 1 (<03:00:00 h), Top 2 (<03:15:00 h), Top 3 (<03:30:00 h). With the aim of comparing the pacing between sex and performance the average speed was normalized. In men, no statistically significant changes were found between performance group and splits. A large number of significant differences between splits and groups were found amongst women: 5–10 km Top 2 vs Top 3 (P = 0.0178), 10–15 km Top1 vs Top 2 (P = 0.0211), 15–20 km Top1 vs Top 2 (P = 0.0382), 20–21.1 km Elite vs Top 2 (P = 0.0129); Elite vs Top 3 (P = 0.0020); Top1 vs Top 2 (P = 0.0233); Top 1 vs Top 3 (P = 0.0007), 25–30 km Elite vs Top 2 (P = 0.0273); Elite vs Top 3 (P = 0.0156), 30–35 km Elite vs Top 2 (P = 0.0096); Top 1 vs Top 2 (P = 0.0198); Top2 vs Top3 (P = 0.0069). In men there were little significant differences based on athletes' performance which implied a similar pacing behaviour. Women presented numerous differences based on their performance which suggested different pacing behaviours.

## Introduction

The Marathon (42195 m) is one of the most participative endurance competitions [1]. With the aim of achieving the race goal it will be mandatory to train different physiological, biomechanical and psychological factors. Also, it will be necessary to establish a specific race pace (RP) which will let the runners maximize their performance through an optimal energy expenditure.

https://www.bmw-berlin-marathon.com/en/impressions/statistics-and-history/results-archive/

**Funding:** The authors received no specific funding for this work.

**Competing interests:** The authors have declared that no competing interests exist.

Within these physiological, biomechanical and psychological factors that influence the RP, it has been demonstrated that thermoregulation [2, 3], glycogen stores depletion [4, 5], neuro-muscular fatigue [6] and the increase of the Rate of Perceived Exertion (RPE) [7, 8], are factors that affect RP directly.

At the same time, variables such as age and runners' training experience have been implicated in the determination of pacing behaviour [9]. Both experience and previous training degree could be decisive factors when developing a RP with little intensity variations during the whole marathon [7]. Besides, it needs to be borne in mind the necessity of continuous decision-making related to competition RP, which will respond to athlete's momentary situation, behaviour of other runners, and environmental conditions [10–12]. Furthermore, the development of RP might be influenced by "stochastic" variables which the runner will not be able to control beforehand. Factors such as race profile [13], temperature and atmospheric humidity [3] play a key role on the race day and hence, in the athlete´s performance.

With the objective of considering all the possible variables which affect the performance (i.e. avoiding premature fatigue due to a race beginning too fast), the election of an optimal RP will be an essential component to improve performance during the race [14]. The different competitive strategies during marathon have been object of attention in several studies based on RP to stablish unique profiles [15–17]. Abiss and Laursen [13] described three main discernible profiles among athletes in endurance and ultra-endurance competitions (>4 h): "negative" characterized by starting under speed average and in the second part of the race increase the speed above the average, "positive" starting clearly beyond the average speed and in the second part of race undergo a dramatic drop in the average speed, and "even pace" which is characterized by a maintenance of steady speed throughout the race. Even though the Positive profile (PP) is the most common strategy among runners [9, 18, 19], in the last 50 years a tendency towards a Negative profile (NP) to be the main one among the best marathon runners has been noticed [20]. Similarly, Breen et al. [9] verified that the best master runners (independently of their age and sex) were characterized by a smaller variation of RP, close to the Even profile (EP).

This ability of the best runners to avoid a large speed reduction in the second part of the competition is based on a greater capacity to adapt oneself as a result of a big load of continuous training [21–23], and the ability to maintain the pace throughout the whole race [10]. Likewise, recreational runners with a largely PP [19, 24] who were able to maintain a steadier RP during a marathon [7, 25] reported higher training volume compared to those with the same training frequency. Therefore, more than the marathon level itself, it is possible that the training volume and runners' former training experience may be more important in determining to develop a negative or EP [19], which will assist a better performance.

At the same time, runner's sex could be related to a greater tendency to develop RP variations. Various studies have corroborated that men tend to reduce RP in the second half of the marathon, corresponding to a PP [3, 9, 26]. On the contrary, when the same situation happens among women, it appears later on and their RP has less variations [17, 24, 26]. In this sense, a study by March et al. [18] underlined the tendency of the female runners of the midwestern U. S. (2005–2007) marathon to adopt profiles which had less rate variation than male runners. This difference might be the result of physiological factors (i.e. higher percentage of slow-twitch oxidative, larger capacity to oxidize more fatty acids and less carbohydrate). Nevertheless, another aspect to bear in mind is the difference in the decision-displaying depending on the sex, showing a major trend to assume a "riskier" profile in men than women, and thus making a faster first half of the marathon [26].

Despite the crucial importance of correctly choosing a competitive profile according to the characteristics of the athlete, which will enable a superior performance in the race, there is a lack of unanimity about what type of profile should be chosen based on the athlete's

performance and sex. Furthermore, notwithstanding this relevance, no previous studies have analysed both factors in elite marathon runners. Thus, the aims of this study were: 1) to analyse the different race profiles displayed in the same marathon based on athlete's performance, and 2) to determine whether significant differences in each race split and the runner's performance implied different race profiles.

## Methods

### Participants

2295 runners were included in the research of which 637 were female. The selection criteria for inclusion in the study were: 1) to have completed the competition; 2) to have successfully completed and recorded the split of each competition; 3) the absence of atypical record; 4) to have finished the race in time ≤03:00:00 h for men and ≤03:30:00 for women.

Splits and race final times of the marathon (Berlin 2017) were obtained through the official web page of Berlin´s Marathon (www.bmw-berlin-marathon.com, 2017) where a total of 39225 individuals participated. After applying all the inclusion criteria, the total number of valid records dropped to 2295 (5.85% of the total runners).

The participants were divided into 4 groups in accordance with their race performance. All athletes which run below or equal to the qualifying standard of the World Athletics Championship (London 2017) established by IAAF (www.iaaf.org, 2017) were included in the Elite group. The subsequent groups were based on 15 minutes time intervals (Top 2 and Top 3), except for Top 1 group which was fixed in 11 minutes (Table 1).

Ten race splits were analysed (0–5 km, 5–10 km, 10–15 km, 15–20 km, 20–21.1 km, 21.1–25 km, 25–30 km, 30–35 km, 35–40 km, 40–42.2 km). Due to the fact that the data was freely available in the public domain, the requirement for informed consent was not necessary. Moreover, this study was conducted in accordance with Helsinki Declaration (1964 and amended in 2013) concerning human experimentation.

### Profile criteria

In order to establish an objective criterion to define the pacing profile (based on RP variation), the present study was based on the method described by Deaner, Carter, Joyner and Hunter [26]. The variation was calculated as (% change = [1 - (second half time—first half time) / first half time] • 100). If the change was less than 10%, be it positive or negative, it was considered EP. If the variation was greater than 10% negative, it was considered PP and if the variation was greater than 10% positive, it was considered NP.

### Statistical analysis

The average speed was normalized for each race split according to the mean speed of the marathon for each individual athlete. Thus, a value higher than 1.00 implies that the split time was faster than

**Table 1. Distribution of groups based on sex and performance.**

| Group | Men | | Women | |
|---|---|---|---|---|
| | Time (h:min:sec) | N | Time (h:min:sec) | N |
| Elite | ≤2h:19:00 | 33 | ≤ 2h:45:00 | 22 |
| Top1 | ≤2h:30:00 | 79 | ≤3h:00:00 | 67 |
| Top2 | ≤2h:45:00 | 384 | ≤3h:15:00 | 159 |
| Top3 | ≤3h:00:00 | 1162 | ≤3h:30:00 | 389 |

the average RP. At the same time, a value below 1.00 means a slower pace than the average race speed. This normalization enables the comparison of race profile between runners, regardless of their performance. Statistical analyses were performed with SAS software (Cary, NC, USA).

For the comparison of the results, a homogeneity of variance test was carried out using the Kolmogorov-Smirnov, Cramer-von Mises and Anderson-Darling tests for all the variables of the study separated by sex and classes. In this way, if the variables were not normally distributed, the Krustal-Wallis test was used (nonparametric ANOVA). A post-hoc comparison with adjustments with multiple comparison tests of two paired sides (Dwass Method, Steel, Critchlow-Fligner) and empirical distribution function test (Kolmogorov-Smirnov, Cramer-von Mises), were undergone using the Group as a classification variable. Cohen's d was used as a measure of effect size, using the reference values of small ($d = 0.2$), medium ($d = 0.5$) and large ($d = 0.8$) for interpreting them as suggested by Cohen [27]. In addition, qualitative analysis of confidence intervals of differences was performed. The significant difference was established for a p value $<0.05$.

## Results

### Sex performance differences in pacing

The normalized speed in each split, sex and performance group is shown in Fig 1 for men and Fig 2 for women. In men, no statistically significant differences were found between groups and splits, except for 0–5 km and 21.1–25 km splits (Table 2).

Amongst women, we observed a large number of differences between groups which were statistically significant (5–10 km, 10–15 km, 15–20 km, 20–21.1 km, 25–30 km and 30–35 km) as seen in Table 3.

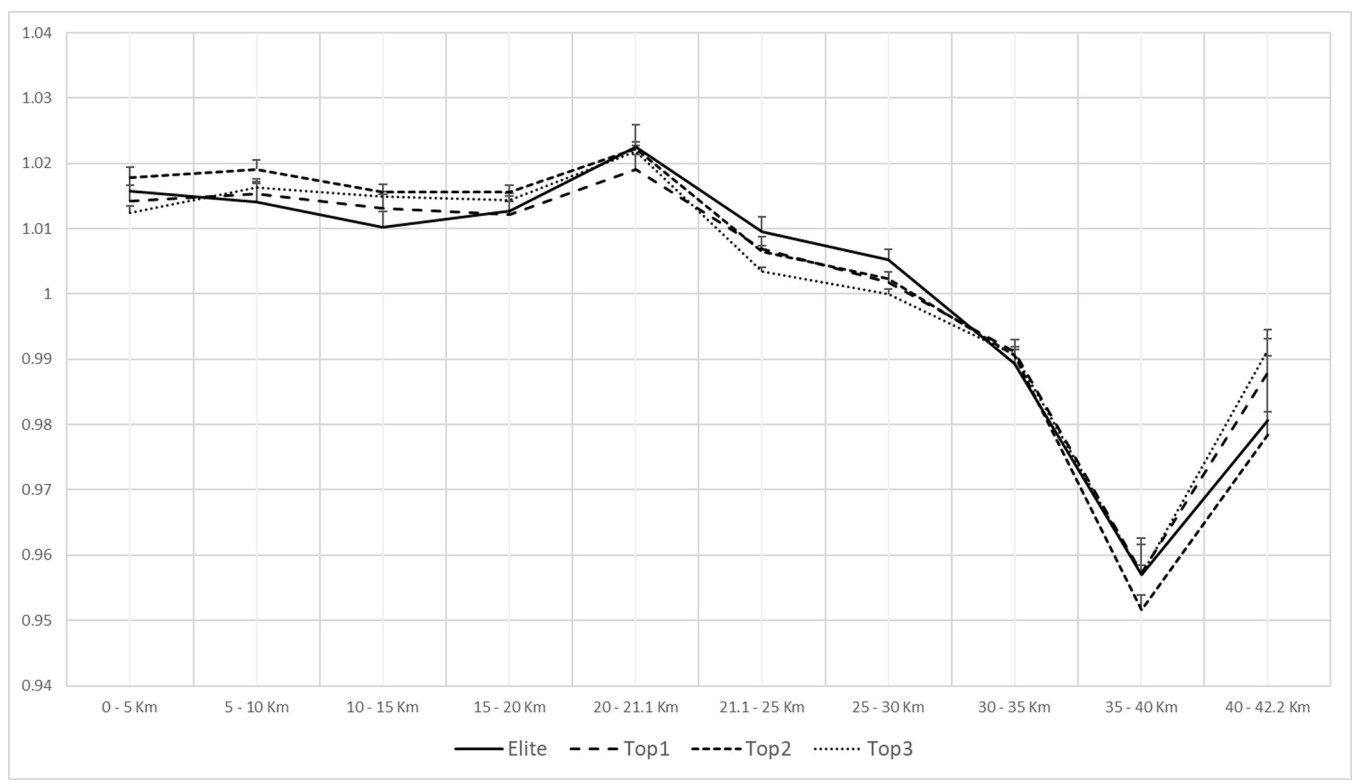

**Fig 1. Paired profiles of average pace per split and group for men.**

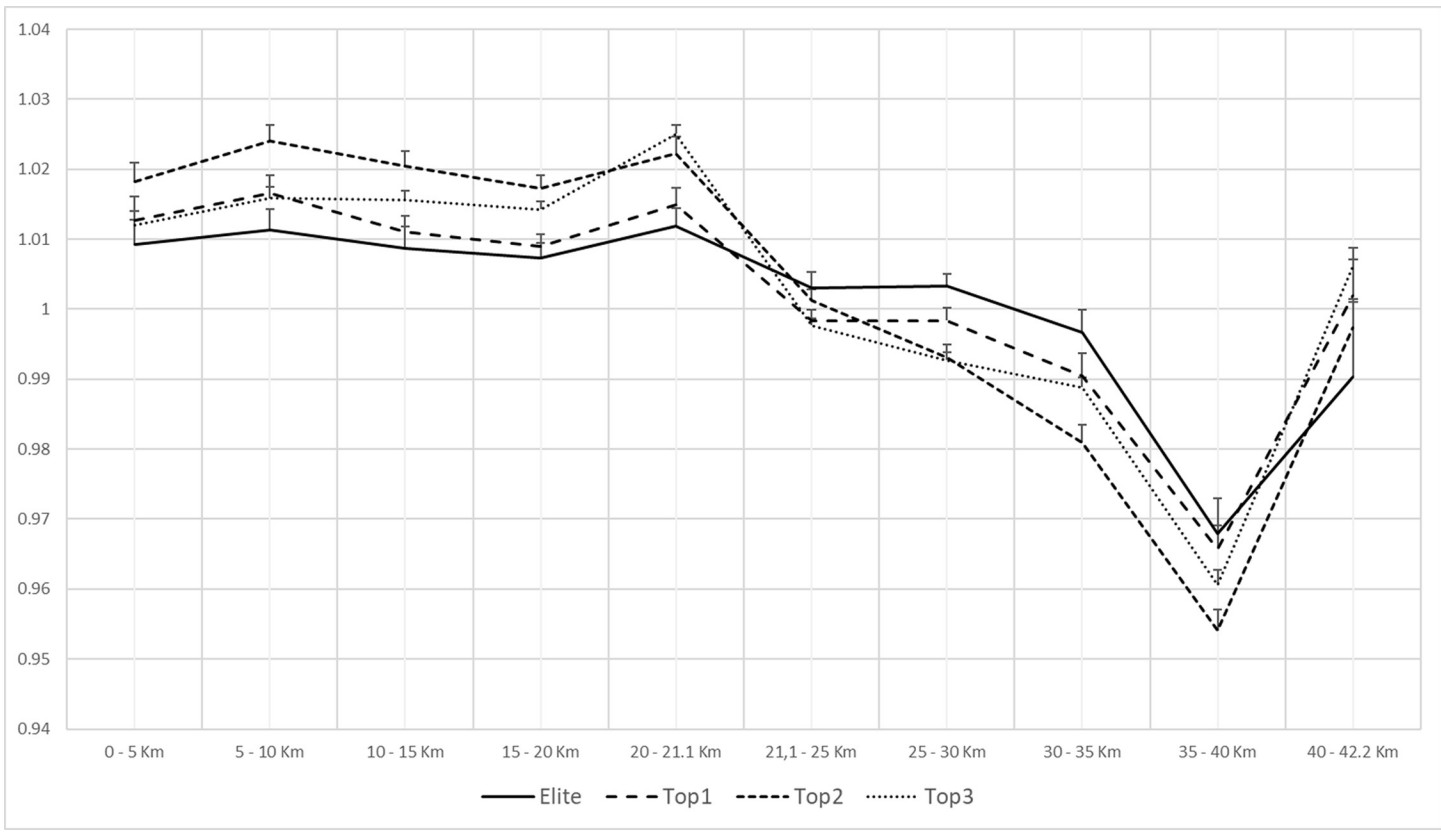

**Fig 2. Paired profiles of average pace per split and group for women.**

Fig 3 represents a comparison between male runners' group and the splits within which significant differences were found. The cumulative distribution function was used to establish how to approach the race within a group. As shown in Fig 3A and 3B Top 3 group was characterized by a wide diversity of speed (from 65% to 120% of average speed). On the contrary, in Elite and Top 1 groups their cumulative distribution was almost vertical (Fig 3A and 3B), which implies that almost 100% of the athletes of these groups were running at their average speed during the marathon.

The subsequent post-hoc analysis presented significant differences in the 0–5 km and 21.1–25 km splits, for Top 2 and Top 3 groups (Table 2). However, the effect size of these differences was small (Table 2). A positive Z score (Table 2) implies that Top 2 group displayed a higher speed relative to mean overall race speed during the marathon than Top 3 for these splits.

On the other hand, women displayed a greater heterogeneity in their start RP (Fig 2). Significant differences were found between groups and their average RP in numerous splits. At the same time, and with the exception of Elite group, Top 1, Top 2 and Top 3 groups were characterized by completing the first half marathon faster than their mean RP (Fig 4A, 4B, 4C, 4d, 4e and 4f) which

**Table 2. Relationship between groups and average pace of competition in men.**

| Split | Group | Z | P | CI (95%) | d |
|-------|-------|-----|-----|---------|-----|
| 0–5 km | Top2 vs Top3 | 3.4924 | 0.0027 | 0.044–0.275 | 0.16 |
| 21.1–25 km | Top2 vs Top3 | 3.0058 | 0.0141 | 0.033–0.264 | 0.15 |

**Table 3. Relationship between groups and average pace of competition in women.**

| Split | Group | Z | P | CI (95%) | d |
|---|---|---|---|---|---|
| 5–10 km | Top2 *vs* Top3 | 2.9310 | 0.0178 | -0.021–0.348 | 0.16 |
| 10–15 km | Top1 *vs* Top2 | -2.8747 | 0.0211 | -0.668 - -0.093 | 0.38 |
| 15–20 km | Top1 *vs* Top2 | -2.6687 | 0.0382 | -0.694 - -0.118 | 0.41 |
| 20–21.1km | Elite *vs* Top2 | -3.0347 | 0.0129 | -0.801 - -0.094 | 0.36 |
| | Elite *vs* Top3 | -3.5734 | 0.002 | -0,955 - -0.093 | 0.53 |
| | Top1 *vs* Top2 | -2.8413 | 0.0233 | -0,548–0.025 | 0.26 |
| | Top1 *vs* Top3 | -3.8579 | 0.0007 | -0,668 --0.147 | 0.41 |
| 25–30 km | Elite *vs* Top2 | 2.7873 | 0.0273 | 0.022–0.919 | 0.47 |
| | Elite *vs* Top3 | 2.9739 | 0.0156 | 0.036–0.898 | 0.47 |
| 30–35 km | Elite *vs* Top2 | 3.1259 | 0.0096 | 0.061–0.959 | 0.51 |
| | Top1 *vs* Top2 | 2.8959 | 0.0198 | 0.024–0.598 | 0.31 |
| | Top2 *vs* Top3 | -3.2265 | 0.0069 | -0.450- -0.080 | 0.27 |

implied, at the same time, a reduction in speed below average RP in the second half of the race. The post-hoc analyses showed significant differences between groups and splits (Table 3).

In contrast with men, significant differences were detected in female runners from 20 km and subsequent splits (until 35–40 km) between groups which suggests a possible difference in competitive profile (Table 3).

## Objective profile per sex performance differences

Tables 4 and 5 report, based on the classification methodology used that the EP was the predominant one in all performance and sex groups, followed by the PP. Only 0.12% of men and 0.16% of women developed the NP.

## Discussion

This study investigated the change in RP during the different splits in Berlin's Marathon (2017 edition) in elite and well-trained men and women. The most important findings were: 1) we observed no difference of RP between performance group for men, which may be related to a

### a Split 0-5 km

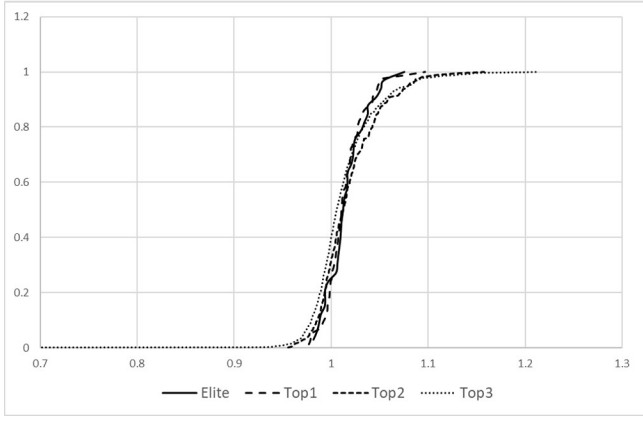

### b Split 5-10 km

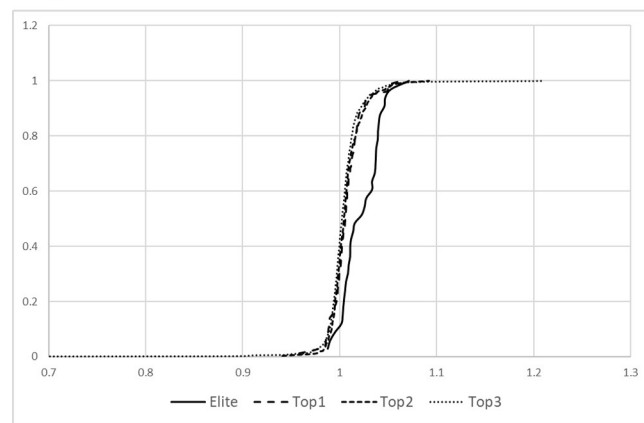

**Fig 3. Competition profile of the male athletes in splits with significant differences between groups.** "X" axis: standardized velocity; "Y" axis: relative frequency. A standardized velocity of "1" represent the mean velocity of the athletes in the marathon.

a Split 5-10 km

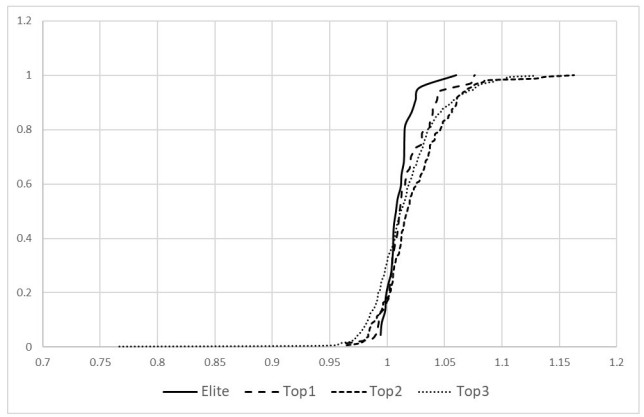

b Split 10-15 km

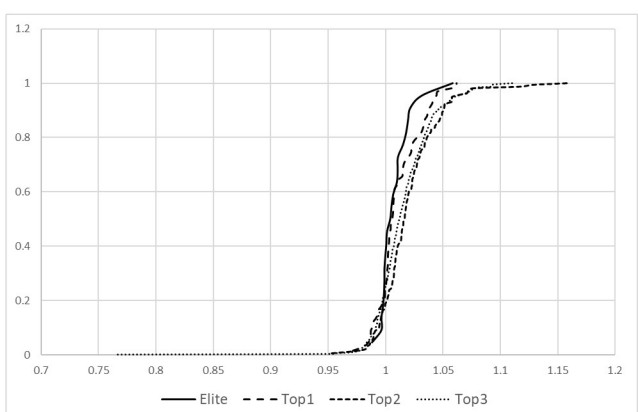

c Split 15-20 km

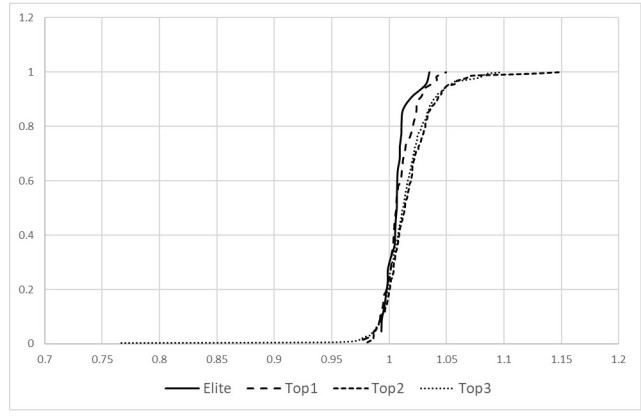

d Split 20-25 km

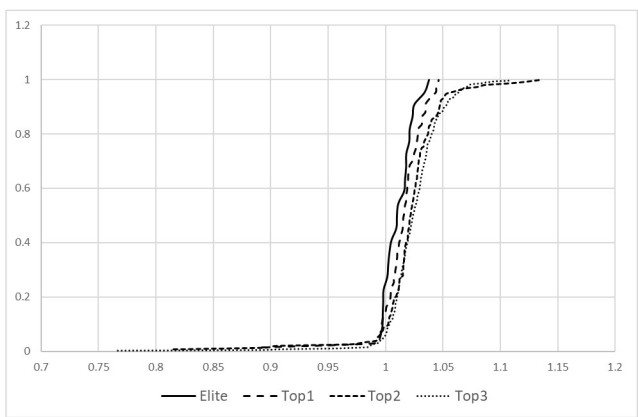

e Split 25-30 km

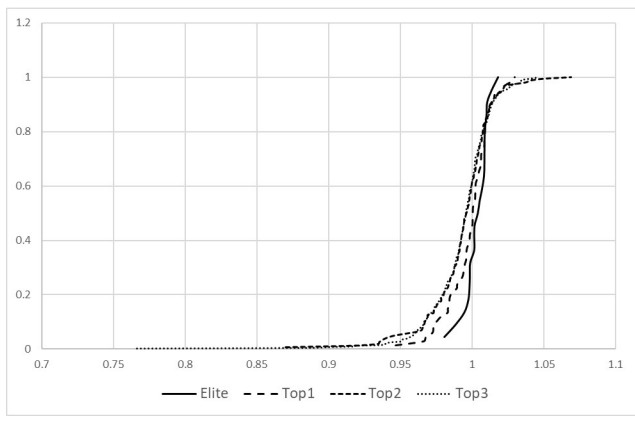

f Split 30-35 km

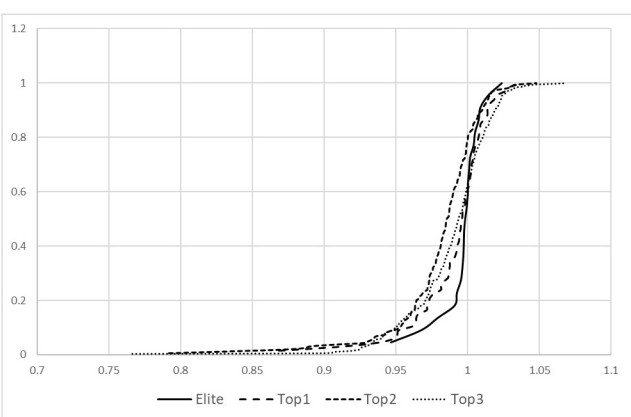

**Fig 4. Competition profile of the female athletes in splits with significant differences between groups.** "X" axis: standardized velocity; "Y" axis: relative frequency. A standardized velocity of "1" represent the mean velocity of the athletes in the marathon.

Table 4. Relation between pacing profiles and athletes'group for men.

| Group | Even Pace | Positive Pace | Negative Pace |
|-------|-----------|---------------|---------------|
| Elite | 96.97% | 3.03% | 0.00% |
| Top1 | 94.94% | 5.06% | 0.00% |
| Top2 | 90.10% | 9.90% | 0.00% |
| Top3 | 92.08% | 7.75% | 0.17% |

similar competition profile; women presented numerous differences among groups, which indicated different competition profiles; 2) the objective profile criteria that was used demonstrated a predominance of the EP for men and women, independent of their performance or RP differences.

## Sex performance differences in pacing

No statistically significant differences were found in men between the decrease in pace below the average speed, the segment in which it was produced and the performance of the runners. Only statistically significant differences were found in the segments 0–5 and 20–21.1 km between the Top2 and Top3 groups.

This lack of differences between splits in male runners, in spite of their performance group, was associated with the same RP. Even though, the runners' performance in this study corresponds to high level athletes, the main profile was the positive one. This data does not differ from previous studies undertaken with lower level marathon runners [9, 18, 19, 28], where the PP is the leading one. A previous study done by Nikolaidis and Knetchtle [19] with a similar performance level (<3:00:00h) did not achieve the same results, with the EP being the most common profile amongst the best runners. It is important to emphasise that a different course topography can influence observed pacing behaviours, especially at the beginning of the competition (i.e. New York *vs* Berlin' marathon), can have an effect on a more conservative start due to race elevation, causing an EP or NP.

In women, several differences were found between performance groups per split. The women Elite group was able to maintain a RP above average RP until the 25–30 km split. These data agree with a previous study of Santos-Lozano and colleagues [17] in the New York Marathon where the fastest female runners experienced a considerable drop in their RP at 30–35 km split. In the same line, Top 1 group tends to develop a similar profile as the Elite group. However, Top 1 group experienced a RP drop earlier in the race (20–21,1 km split) but less pronounced. These data are in accordance with previous studies which highlighted the best marathon runners were prone to display less variations in their RP [15, 16, 26, 29]. On the other hand, as the runners' performance declined (Top 2 and 3 groups), the drop in their RP was larger, with a tendency to develop a PP. Deaner and colleagues [26] showed a similar trend with female heterogeneous performance runners (<3 h:22 min >5 h:36 min). Even though the tendency of increasing the development of PP is linked to a low level of

Table 5. Relation between pacing profiles and athletes'group for women.

| Group | Even Pace | Positive Pace | Negative Pace |
|-------|-----------|---------------|---------------|
| Elite | 95.45% | 4.55% | 0.00% |
| Top1 | 95.52% | 4.48% | 0.00% |
| Top2 | 90.57% | 9.43% | 0.00% |
| Top3 | 91.26% | 8.48% | 0.26% |

performance, women are more prone to develop less variations in the RP in a marathon than men [26, 29].

## Objective profile per sex performance differences

The present study, after applying an objective profile criterion [26], showed a majority of EP (>90%), independently of sex and performance differences. Previous studies do not coincide with the results obtained in this work regarding the type of the most common profile (positive) for men in races over this distance [17, 30, 31]. This "classic" profile is characterized by a dramatic drop in the average speed of the race during the second part of the competition [20], which is related to the depletion of glycogen stores muscle [4], an increase in internal temperature and increase in RPE [8] among other physiological, biomechanical and psychological factors. In addition, it should be borne in mind that this type of competition profile may respond to an unrealistic perception of the athlete in relation to his ability to maintain speed throughout the race [13]. For all these reasons Díaz et al. [20] propose to aim for NP as an optimal strategy.

On the other hand, the EP, which is predominant in this study although more common in shorter duration competitions [32], will be able to report a better performance. This will be achieved thanks to lower fluctuations in the race speed, which lead to a better energy efficiency [14]. The finding in the present study of a wide majority of runners display EP (independent of their sex and performance) does not coincide with previous studies which underlined a lack of significant variation of RP (related to Negative and EP) just in some of the best runners [9, 19, 28].

Regarding the NP, a study by Renfree and Gibson [16] showed the predominance of this type of profile in the athletes with greater performance in the first segment (0–5 km) during the women's marathon world championship (2009). Thus, this group adopted a NP, during the first 10 km, they decreased its RP later than the rest, and therefore obtained a higher performance. In the present study, only 0.12% of men and 0.16% of women displayed this type of profile. Based on the data obtained in this study, it can be observed that this type of profile is minority regardless of sex and performance.

However, the detection of a small percentage of athletes who developed a NP and PP in this study can be explained by the application of the classification criteria proposed by Deaner et al. [26]. If we exclusively follow the criteria set by Abbiss and Laursen [13], observing Figs 1 and 2, we could interpret the existence of a majority PP in both men and women, regardless of their final performance. The graphic representation (Figs 1 and 2) and the detection of the main competitive profiles in an objective manner by the proposal of Deaner et al. [26] do not have a coherent relationship.

One of the possible explanations for the difference between the data obtained in this study, with respect to main competitive profile, and others [17, 30, 31], may be based on the low sensitivity of the method used to establish a belonging to each profile. The methodology used by Deaner et al. [26], which identifies differences <10% between the first and second half as EP, means that there might be a considerable difference between the two halves but the developed profile cannot be differentiated. By means of such wide margins of time the type II error (β) can be very high. In the same way, for a greater precision and detection of developed profile it will be necessary to establish a greater number of segments to be assessed.

The need for a more sensitive methodology becomes more relevant in women due to the large number of differences between the performance group and the average speed of each segment. The method used to detect the different profiles [26] shows that all groups had the EP as predominant profile (Table 5). However, despite the fact that in all the female runner's groups

this profile similarity was detected, significant differences were found between the normalized speed in numerous segments based on the group of belonging, which suggests different profiles.

Finally, only several studies have proposed a random percentage of decrease in the average speed to perform a classification [26, 33]. In this way, PP and NPs are clearly defined. However, there is a lack of studies which establishes the maximum percentage of variation between segments to be considered uniform profile. Recently, a work by Diaz et al. [34] explained an objective method to categorize the main three profiles in marathon runners. Nevertheless, future researches will have to be conducted to promote an objective classification capable of detecting more than these three profiles, and thus increase the potential number of profiles and unify classification criteria.

## Limitations

The present study has some limitations. One of them is the length of each split (5 km, except 20–21.1 km and 40–42.2 km splits). If shorter splits had been used (i.e. 1 km), we may have found greater variability [24, 35]. Another limitation of this study is that we did not consider the effect of running in a group (packing) into the pacing strategy, and this could be a key point which influences pacing strategy [15]. The faster the runner's RP is, the less participants running together and when one of them dropped out from his/her group their RP, and probably pacing strategy, change making a drop in the RP and a race profile change more probable. Conversely, the lower level runners may have benefitted from a greater density of runners during the race, which could help to maintain for a longer period of time a steady RP. Another limitation of this study is that it was not considered that the lower the performance level of runners is (in these crowed marathons), the more likely they will suffer from "runner's jam" at the start of the race, which might change their RP in the first kilometres and the subsequent race profile. Unfortunately, our study has not been capable of detecting this as the marathon races are influenced by plenty of factors, making them complex systems [10].

## Conclusion

The present study demonstrates that no differences exist in pacing among elite ($\leq$2h:19:00) and well-trained ($\leq$3h:00:00) male marathoners, which means they display similar race profiles despite their performance. However, in women, there were numerous significant differences based on their performance and competitive segment, which indicates different competitive profiles. The best female runners ($\leq$3:00:00 h) developed less RP variation than lower performance groups. This performance difference among runner groups could imply the use of different race profiles.

## Acknowledgments

The authors of the present study want to express their gratitude to Guillermo Chávarri from RUNNEA for his help in the acquisition and processing of data.

## Author Contributions

**Conceptualization:** Iker Muñoz-Pérez, Marcos Mecías-Calvo, Jorge Crespo-Álvarez.

**Data curation:** Iker Muñoz-Pérez, Marcos Mecías-Calvo, Pablo Agudo-Toyos, Carlos Lago-Fuentes.

**Formal analysis:** Jorge Crespo-Álvarez, María Luisa Sámano-Celorio.

**Investigation:** Iker Muñoz-Pérez, Marcos Mecías-Calvo, Jorge Crespo-Álvarez, Pablo Agudo-Toyos, Carlos Lago-Fuentes.

**Methodology:** Iker Muñoz-Pérez, Marcos Mecías-Calvo, Jorge Crespo-Álvarez, Pablo Agudo-Toyos, Carlos Lago-Fuentes.

**Project administration:** Iker Muñoz-Pérez, Marcos Mecías-Calvo, Carlos Lago-Fuentes.

**Resources:** Iker Muñoz-Pérez, Pablo Agudo-Toyos.

**Supervision:** Iker Muñoz-Pérez, Marcos Mecías-Calvo, María Luisa Sámano-Celorio.

**Validation:** Iker Muñoz-Pérez, Marcos Mecías-Calvo, Jorge Crespo-Álvarez, María Luisa Sámano-Celorio, Pablo Agudo-Toyos, Carlos Lago-Fuentes.

**Visualization:** Iker Muñoz-Pérez, Marcos Mecías-Calvo, Jorge Crespo-Álvarez, María Luisa Sámano-Celorio, Pablo Agudo-Toyos, Carlos Lago-Fuentes.

**Writing – original draft:** Iker Muñoz-Pérez, Marcos Mecías-Calvo, Jorge Crespo-Álvarez, María Luisa Sámano-Celorio, Pablo Agudo-Toyos, Carlos Lago-Fuentes.

**Writing – review & editing:** Iker Muñoz-Pérez, Marcos Mecías-Calvo, Jorge Crespo-Álvarez, María Luisa Sámano-Celorio, Pablo Agudo-Toyos, Carlos Lago-Fuentes.

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
