## [Decision Letter · Decision Letter 0]

14 Apr 2020

PONE-D-20-04748

Different Race Paces and Race Profiles in the best runners on Berlin´s Marathon

PLOS ONE

Dear Dr Mecías Calvo,

Thank you for submitting your manuscript to PLOS ONE. After careful consideration, we feel that it has merit but does not fully meet PLOS ONE’s publication criteria as it currently stands. Therefore, we invite you to submit a revised version of the manuscript that addresses the points raised during the review process.

While the referees have scored low this manuscript, I would expect it may be considered when all the concerns raised by the referees be addressed in another round of reviews. Importantly, an English revision by a mother tongue scientist is required before further considerations. Therefore, this Editor would like to give a chance to the authors but clearly stating that this does not necessarily means that the manuscript was finally accepted. The major reason for this is the novelty of the results.

We would appreciate receiving your revised manuscript by May 29 2020 11:59PM. To enhance the reproducibility of your results, we recommend that if applicable you deposit your laboratory protocols in protocols.io, where a protocol can be assigned its own identifier (DOI) such that it can be cited independently in the future. For instructions see: http://journals.plos.org/plosone/s/submission-guidelines#loc-laboratory-protocols

We look forward to receiving your revised manuscript.

Kind regards,

Daniel Boullosa

Academic Editor

PLOS ONE

2. Please ensure you have thoroughly discussed any potential limitations of this study within the Discussion section.

3. Please ensure that you include a title page within your main document. We do appreciate that you have a title page document uploaded as a separate file, however, as per our author guidelines (http://journals.plos.org/plosone/s/submission-guidelines#loc-title-page) we do require this to be part of the manuscript file itself and not uploaded separately.

Reviewers' comments:

Reviewer's Responses to Questions

**Comments to the Author**

1. Is the manuscript technically sound, and do the data support the conclusions?

Reviewer #1: Yes

Reviewer #2: Partly

2. Has the statistical analysis been performed appropriately and rigorously? 

Reviewer #1: Yes

Reviewer #2: Yes

3. Have the authors made all data underlying the findings in their manuscript fully available?

Reviewer #1: Yes

Reviewer #2: No

4. Is the manuscript presented in an intelligible fashion and written in standard English?

Reviewer #1: No

Reviewer #2: No

5. Review Comments to the Author

Reviewer #1: This study analyzed pacing behaviours displayed by 2295 runners in the 2017 Berlin marathon. The primary finding was that after speeds were normalised to race average, there were few differences between athletes of different absolute performance levels in males, but in females there were a large number of differences between groups at various points.

Unfortunately, as it stands, I feel the quality of English language used is insufficient to allow publication of the paper in its current format as there are numerous grammatical errors throughout. The work would greatly benefit from detailed proofreading , as there are a lot of places when the meaning of statements is unclear.

Furthermore (although the findings are slightly surprising & differ from many other analyses in the literature) I dont really see what is novel about this study or what it brings to the body of literature on pacing. The rationale is quite 'shaky' and you cant determine what stategy an athlete 'should' adopt based on a descriptive study of one race such as this.

Methodology is generally sound, but the groupings of athletes seems arbitrary and requires justification. for example the >3 hours group in males potentially covers a very wide range of abilities who may have widely varying motivational profiles.

Reviewer #2: The current study examines the pacing strategies conducted by a great number of runners competing at 2017 Berlin of Marathon and compare these strategies by level of performance. They also describe separately the different profiles observed in males and females. This study is interesting given the great number of runners nowadays who train for running marathon races and I also find interesting the specific comparison made between pacing strategies conducted by elite and recreational runners.

However, I have major concerns which should be addressed by authors prior to consider myself the present manuscript suitable for publication. On the one hand, the manuscript contains several grammar mistakes throughout which should be corrected. I encouraged authors to ask for the help of a native English speaker to conduct these corrections. On the other hand, I suggest authors to distribute differently the narrative structure which may explain the findings obtained in a more thorough way.

Specific comments are raised below.

Title:

I suggest a more correct (grammar) and precise title: ‘Different race pacing strategies among runners covering the 2017 Berlin Marathon under 3 hours and 30 minutes’

Abstract:

No reference to the specific race from which data was taken was made. I also miss some descriptions regarding the specific pacing profiles conducted by both women and men. No reference was made to the data normalization process conducted prior to conducting the different comparisons. This is a critical aspect which should be highlighted. I think that removing some of the specific results found or summarizing them through the indication of ranges rather is needed to include these necessary points suggested.

Introduction:

Authors failed to structure adequately the introduction. In the current study the influence of two main variables (i,e., sex and level of performance) on performance outcomes was measured. Therefore, the introduction should focus on explaining the existing evidence describing the typical pacing strategies observed in elite runners and runners who possesses lower performance level as well as the differences observed by sex. Results from current evidence should be explained according to these aspects.

Methods:

Results:

The table 3 is not clearly indicated. For example, it is not clear which comparisons belong to each split.

Figure 3 and 4 are difficult be understood. For example, no reference to the meaning of X and Y axes is indicated in the legend. It should be clarified.

Discussion:

From L141 to L165 authors just describe the results found without discussing them according to existing research. This should be part of the results rather than the discussion.

Referencing swimming studies is not worth in the discussion of a pacing analysis in marathon given the amount of existing pacing studies focused specifically on the marathon race.

According to the structure suggested for the introduction, authors are encouraged to compare the outcomes observed in the current study separately. For example, comparing the pacing profile observed in elite runners with results observed in existing research focused specifically on elite runners. Same should be done with research focused on runners with lower level of performance. Additionnaly, same should be done when comparing observations made in female runners as some previous research just focused on this specific population. In this sense, I miss the reference to studies strongly related to this specific topic which focused on pacing strategies conducted in marathon runners.

In addition, when conducting studies of such type, there are some limitations which should be addressed in the text. For example, the influence of running in a group on the pacing strategy selected was not even mentioned. As long as the level of performance is decreasing, the density of runners representing the number of participants running together is also increasing. For example, it is very likely that a higher proportion of runners belonging to the Elite group dropped out from the group they were running with and had to run individually. As the influence of this type of circumstances was not analyzed, this should be stated as a limitation. Hanley (5) studied the influence of running in a group (packing) in elite marathons. In addition, Renfree and Casado reviewed the different existing influences in Athletic races which transform them in complex systems.

Renfree A., Casado A. (2018) Athletics races represent complex systems and pacing behavior should be viewed as emergent phenomenon. Frontiers in Physiology. https://doi.org/10.3389/fphys.2018.01432

Furthermore, very likely those runners achieving lower level of performance experienced difficulties to run at a given pace in the beginning of the race as in crowdy races such as Berlin Marathon many runners start the race together and they usually get blocked. This also represents a limitation of this study which should be stated specially when referring to the specific comparison made between groups of different level of performance.

6. PLOS authors have the option to publish the peer review history of their article (what does this mean?). If published, this will include your full peer review and any attached files.

Reviewer #1: No

Reviewer #2: No

---

## [Author Response · Author response to Decision Letter 0]

27 May 2020

AUTHORS’ INTRODUCTION

Thank you for providing us with constructive comments regarding our article and for inviting us to submit this revised version. We have made several changes to the manuscript and have addressed all of the comments made by the reviewers. We honestly think that the article has improved as a consequence and is now ready for publication. Thank you very much for your help.

Our reply to the reviewer’s comments can be found below. Our changes in the manuscript are highlighted by track changes.

REVIEWER 1 

Comments for the Author

Unfortunately, as it stands, I feel the quality of English language used is insufficient to allow publication of the paper in its current format as there are numerous grammatical errors throughout. The work would greatly benefit from detailed proofreading, as there are a lot of places when the meaning of statements is unclear. 

Authors’ reply: the English quality of the present manuscript has been improved.

Furthermore (although the findings are slightly surprising & differ from many other analyses in the literature) I dont really see what is novel about this study or what it brings to the body of literature on pacing. 

Authors’ reply: the novelty of our study lies in the fact that there are no differences in the race profile between elite and well-training runners when the RP is normalized. This finding is remarkably different compared with previous studies. In turn, women develop less RP variations, as previous studies have showed. However, there are no differences among the best runners and the subsequent group (Top1). This highlight represents a similar race strategy in the elite and well-trained athletes. Another novel point in our study is the comparison among smaller performance groups than previous studies. This has been afforded us to detect differences for women, and no differences, for men, regarding the RP distribution. To sum up, to the best of the author’s knowledge, these results have not yet been shown in other studies. 

The rationale is quite 'shaky' and you can’t determine what stategy an athlete 'should' adopt based on a descriptive study of one race such as this. 

Authors’ reply: thanks for this feedback. In this sense, the aim of this study has not been to determine the best race strategy based on the runner performance. We completely agree with reviewer’s point of view about this key aspect, it is not possible to decide the best marathon strategy based on a descriptive analysis. 

Methodology is generally sound, but the groupings of athletes seems arbitrary and requires justification. for example the >3 hours group in males potentially covers a very wide range of abilities who may have widely varying motivational profiles. 

Authors’ reply: we based on the qualifying standard of the World Athletics Championship as an objective way to classify the best runners. After that, we established groups every 15 minutes interval (except for Top1 men group). With the aim of ensuring that the proposed classification reflected the differences in performance between groups, a Kruskal-Wallis test, and subsequent post-hoc test with Bonferroni adjustments, was conducted. This analysis showed that all the groups were significantly different among them.

REVIEWER 2 

Comments for the Author

Title:

I suggest a more correct (grammar) and precise title: ‘Different race pacing strategies among runners covering the 2017 Berlin Marathon under 3 hours and 30 minutes’. 

Authors’ reply: thanks for this suggestion. We have rewritten the title, taking into account the reviewer’s advice.

Abstract:

No reference to the specific race from which data was taken was made. 

Authors’ reply: this information has been included in the abstract, lines 4-5. 

I also miss some descriptions regarding the specific pacing profiles conducted by both women and men. 

Authors’ reply: we completely agree with this point. However, due to the limited extension of the abstract, it has not been possible to include this valuable information in this section. 

No reference was made to the data normalization process conducted prior to conducting the different comparisons. This is a critical aspect which should be highlighted. I think that removing some of the specific results found or summarizing them through the indication of ranges rather is needed to include these necessary points suggested. 

Authors’ reply: thanks for this note. We have modified the abstract and completed it with the indicated information (lines 8-9). 

Introduction:

Authors failed to structure adequately the introduction. In the current study the influence of two main variables (i,e., sex and level of performance) on performance outcomes was measured. Therefore, the introduction should focus on explaining the existing evidence describing the typical pacing strategies observed in elite runners and runners who possesses lower performance level as well as the differences observed by sex. Results from current evidence should be explained according to these aspects. 

Authors’ reply: thanks for this feedback. We have improved the introduction, focusing its structure on these two main variables (sex: lines 66 and level of performance: 58).

Methods:

Results:

The table 3 is not clearly indicated. For example, it is not clear which comparisons belong to each split. 

Authors’ reply: thanks for this note. We have improved the information regarding table 3 and its description. 

Figure 3 and 4 are difficult be understood. For example, no reference to the meaning of X and Y axes is indicated in the legend. It should be clarified. 

Authors’ reply: thanks for this note. We have included, in the figure legend, the meaning of “X” and “Y” axes. 

Discussion:

From L141 to L165 authors just describe the results found without discussing them according to existing research. This should be part of the results rather than the discussion. 

Authors’ reply: we agree with reviewer’s point of view, this subsection of discussion has been discarded. 

Referencing swimming studies is not worth in the discussion of a pacing analysis in marathon given the amount of existing pacing studies focused specifically on the marathon race. 

Authors’ reply: thanks for this note. We have dismissed these studies and used more suitable ones.

According to the structure suggested for the introduction, authors are encouraged to compare the outcomes observed in the current study separately. For example, comparing the pacing profile observed in elite runners with results observed in existing research focused specifically on elite runners. Same should be done with research focused on runners with lower level of performance. Additionnaly, same should be done when comparing observations made in female runners as some previous research just focused on this specific population. In this sense, I miss the reference to studies strongly related to this specific topic which focused on pacing strategies conducted in marathon runners. 

Authors’ reply: thanks for this valuable notice. Taking into account this feedback, several modifications have been conducted in discussion section. Two subsections have been created: Sex per performance differences in pacing and objective profile per sex per performance (lines 177-205).

In addition, when conducting studies of such type, there are some limitations which should be addressed in the text. For example, the influence of running in a group on the pacing strategy selected was not even mentioned. As long as the level of performance is decreasing, the density of runners representing the number of participants running together is also increasing. For example, it is very likely that a higher proportion of runners belonging to the Elite group dropped out from the group they were running with and had to run individually. As the influence of this type of circumstances was not analyzed, this should be stated as a limitation. Hanley (5) studied the influence of running in a group (packing) in elite marathons. In addition, Renfree and Casado reviewed the different existing influences in Athletic races which transform them in complex systems.

Renfree A., Casado A. (2018) Athletics races represent complex systems and pacing behavior should be viewed as emergent phenomenon. Frontiers in Physiology. https://doi.org/10.3389/fphys.2018.01432

Furthermore, very likely those runners achieving lower level of performance experienced difficulties to run at a given pace in the beginning of the race as in crowdy races such as Berlin Marathon many runners start the race together and they usually get blocked. This also represents a limitation of this study which should be stated specially when referring to the specific comparison made between groups of different level of performance. 

Authors’ reply: thanks for these valuable suggestions. We have included the limitations section considering these useful advices.

---

## [Decision Letter · Decision Letter 1]

23 Jun 2020

PONE-D-20-04748R1

Different race pacing strategies among runners covering the 2017 Berlin Marathon under 3 hours and 30 minutes

PLOS ONE

Dear Dr. Mecías Calvo,

Thank you for submitting your manuscript to PLOS ONE. After careful consideration, we feel that it has merit but does not fully meet PLOS ONE’s publication criteria as it currently stands. Therefore, we invite you to submit a revised version of the manuscript that addresses the points raised during the review process.

Please, after consideration of the reviewer's requests, ask a mother tongue sport scientist to revise the manuscript.

We look forward to receiving your revised manuscript.

Kind regards,

Daniel Boullosa

Academic Editor

PLOS ONE

Reviewers' comments:

Reviewer's Responses to Questions

**Comments to the Author**

1. If the authors have adequately addressed your comments raised in a previous round of review and you feel that this manuscript is now acceptable for publication, you may indicate that here to bypass the “Comments to the Author” section, enter your conflict of interest statement in the “Confidential to Editor” section, and submit your "Accept" recommendation.

Reviewer #1: (No Response)

Reviewer #2: All comments have been addressed

2. Is the manuscript technically sound, and do the data support the conclusions?

Reviewer #1: Yes

Reviewer #2: Yes

3. Has the statistical analysis been performed appropriately and rigorously? 

Reviewer #1: Yes

Reviewer #2: Yes

4. Have the authors made all data underlying the findings in their manuscript fully available?

Reviewer #1: Yes

Reviewer #2: Yes

5. Is the manuscript presented in an intelligible fashion and written in standard English?

Reviewer #1: No

Reviewer #2: Yes

6. Review Comments to the Author

Reviewer #1: I thank the authors for the revisions to the manuscript. However, there are still numerous grammatical errors that should be corrected before it can be considered suitable for publication. Specifically:

Abstract

Line 15 – change to analyse different pacing behaviours, not ‘race paces’ (which will obviously be higher in faster runners)

Line 22 – change to ‘no statistically significant changes were found…’

Line 23 – change to ‘a large number of significant differences between splits and groups were found amongst women’

Line 29-30. Not clear what you mean

Line 30-32, As above. Do you mean a positive profile was the most commonly displayed one?

Introduction

Line 38 – The marathon is one of the most popular…

Line 41 – delete ‘in the race day’

Line 44 – replace ‘pointed out’ with ‘demonstrated’

Line 47-8 – have been implicated in the determination of pacing behaviour

Line 57 – delete ‘a’

Line 58 – delete ‘the’ before performance

Line 59 – 60: makes little sense

Line 62 – change for to by

Line 64 – change dramatically to dramatic

Line 66- change the runners to ‘runners’

Line 67 – change ‘pf the ‘ to ‘towards a’

Line 71 – change deep speed drop to ‘large speed reduction’

Line 74 – change ‘main’ to ‘largely’

Line 75 – what do you mean by ‘broader’? higher?

Line 77 – ‘may be more important in determining…’

Line 78 – change ease to ‘assist’

Line 81 – change ‘which answers to’ to ‘corresponding to…’

Line 83 – delete ‘made’

Line 89 – change ‘making’ to ‘displaying’

Line 91 – change ‘greater’ to ‘superior’

Line 95 – change ‘race paces’ to race profiles displayed in the same marathon

Method

102 – what do you mean by ‘recorded the split’?

Line 102 – what is an atypical record?

Line 106 – changed ‘records were obtained’ to ‘ individuals participated’

Line 106 – change amount to number

Line 107 - end sentence after info in brackets. The rest is repetition.

Line 109 – ‘into’ 4 groups

Line 112 – ‘were based on 15 minute time intervals’

Line 116-117 – As the data was freely available in the public domain, the requirement for informed consent was not necessary.

Line 119 – amended

Line 122 – change to ‘pacing profile’

Line 124 – delete ‘percentage’

Line 125 – delete ‘percentage’

Line 130 – ‘the marathon for each individual athlete’

Results

Line 146 – delete ‘per’

Line 148 – in men, no statistically significant differences were found…

Line 150 – ‘Amongst women, we observed a large number of differences between groups which were statistically significant’

Line 153 – change men to male, and ‘the spits within which significant differences were found’.

Line 157 – change practically to almost

Line 161-162 – change implied to implies and ‘displayed a higher speed relative to mean overall race speed’

Table 2 – statistically significant but small effect size. Worthy of comment?

Line 165 – change confirmed to displayed

Line 168 – change for to ‘by’ , a to ‘the’, and beyond to ‘faster than’

Line 169 – ‘a reduction in speed below average…’

Line 173 – change corroborates to suggests

Line 174 – change ‘of’ to ‘in’

Line 176 – delete second ‘per’

Line 177-8 – Tables 4 and 5 suggest that EP was the most commonly displayed pacing profile in both genders and at all levels of performance. This was followed by PP.’ Delete ‘regarding this’

Discussion

Line 185 – we observed no differences in RP…

Line 187 – change showed to demonstrated

Line 190 – delete ‘per’

Line 196 – change of to in

Line 198 – change with to from and which to where

Line 199 – ‘a previous study by…’

Line 200 – ‘with EP being the most common profile amongst the best runners’.

Line 201 – ‘it is important to emphasise that a different course topography can influence observed pacing behaviours’

Line 204-205 – delete ‘which involved different competition profile’

Line 205 – the womens elite group was able to maintain a RP above average until …

Line 209 – delete experimented and replace with ‘ experienced a RP drop earlier in the race’

Line 211 – change develop to display

Line 214 - showed ‘a similar…’

Line 217 - in ‘a’ marathon

Line 218 – delete second ‘per’

Line 221 – the most common profile

Line 225 – an increase in RPE

Line 228-9 – Diaz et al propose to aim for EP as an optimal strategy

Line 230-231 – reword this sentence please

Line 233 – ‘that the majority of runners display EP..’

Line 237 – St Clair Gibson

Line 241 – change developed to displayed

Line 246 – delete ‘because’

Line 250 – you need to explain this statement

Line 253 – is there a word missing after ‘establish’?

Line 258 – assessed is probably a better word than valued here

Line 270 – replace ‘on’ with ‘in’

Line 276 – If shorter splits had been used, we may have found greater variability

Line 277-8 – ‘we did not consider the effect of running in a group’

Line 281 - ‘variated’? I do not know this word

Line 282 – change inversely to conversely and ‘may have benefitted from a greater density of runners…’

Line 291 – ‘well-trained’ and change develop to display

Line 292 – delete ‘of’

Line 293 – replace ‘points out’ with ‘indicates’

Reviewer #2: The authors have now improved the quality of the present manuscript substantially. Congratulations for the great work.

7. PLOS authors have the option to publish the peer review history of their article (what does this mean?). If published, this will include your full peer review and any attached files.

Reviewer #1: No

Reviewer #2: Yes: Arturo Casado

---

## [Author Response · Author response to Decision Letter 1]

8 Jul 2020

AUTHORS’ INTRODUCTION

Thank you for providing us with constructive comments regarding our article and for inviting us to submit this revised version. We have made several changes to the manuscript and have addressed all of the comments made by the reviewer. We honestly think that the article has improved as a consequence and is now ready for publication. Thank you very much for your help.

Our reply to the reviewer’s comments can be found below. Our changes in the manuscript are highlighted by track changes.

REVIEWER 1 

Comments for the Author

Abstract:

Line 15 – change to analyse different pacing behaviours, not ‘race paces’ (which will obviously be higher in faster runners). The changes that the reviewer suggests have been made.

Line 22 – change to ‘no statistically significant changes were found…’ The change that the reviewer suggests has been made.

Line 23 – change to ‘a large number of significant differences between splits and groups were found amongst women’. The change that the reviewer suggests has been made.

Line 29-30. Not clear what you mean. We decided to remove the last part of the sentence to improve the understanding.

Line 30-32, As above. Do you mean a positive profile was the most commonly displayed one? We decided to remove the last part of the sentence to improve the understanding.

Introduction:

Line 38 – The marathon is one of the most popular… The change that the reviewer suggests has been made.

Line 41 – delete ‘in the race day’ The change that the reviewer suggests has been made.

Line 44 – replace ‘pointed out’ with ‘demonstrated’ The change that the reviewer suggests has been made.

Line 47-8 – have been implicated in the determination of pacing behavior. The change that the reviewer suggests has been made.

Line 57 – delete ‘a’. The change that the reviewer suggests has been made.

Line 58 – delete ‘the’ before performance. The change that the reviewer suggests has been made.

Line 59 – 60: makes little sense. The sentence has been modified to improve the understanding

Line 62 – change for to by. The change that the reviewer suggests has been made.

Line 64 – change dramatically to dramatic. The change that the reviewer suggests has been made.

Line 66- change the runners to ‘runners’. The change that the reviewer suggests has been made.

Line 67 – change ‘pf the ‘ to ‘towards a’. The change that the reviewer suggests has been made.

Line 71 – change deep speed drop to ‘large speed reduction’. The change that the reviewer suggests has been made.

Line 74 – change ‘main’ to ‘largely’. The change that the reviewer suggests has been made.

Line 75 – what do you mean by ‘broader’? higher? The change that the reviewer suggests has been made.

Line 77 – ‘may be more important in determining…’ The change that the reviewer suggests has made.

Line 78 – change ease to ‘assist’. The change that the reviewer suggests has been made.

Line 81 – change ‘which answers to’ to ‘corresponding to…’ The change that the reviewer suggests has been made.

Line 83 – delete ‘made’. The change that the reviewer suggests has been made.

Line 89 – change ‘making’ to ‘displaying’. The change that the reviewer suggests has been made.

Line 91 – change ‘greater’ to ‘superior’. The change that the reviewer suggests has been made.

Line 95 – change ‘race paces’ to race profiles displayed in the same marathon. The change that the reviewer suggests has been made.

Method

102 – what do you mean by ‘recorded the split’? Sometimes the 5 km splits are not recorded and the organization of the race publishes partial numbers of splits but not all of them. In the present study if a runner did not have every split recorded he/she was excluded from the data analyze. 

Line 102 – what is an atypical record? Sometimes the record of the data is wrong. For example, for 5-10 km split it could appear a runner ran at 250% of his average speed, and this is practically impossible. 

Line 106 – changed ‘records were obtained’ to ‘ individuals participated’. The change that the reviewer suggests has been made.

Line 106 – change amount to number. The change that the reviewer suggests has been made.

Line 107 - end sentence after info in brackets. The rest is repetition. The change that the reviewer suggests has been made.

Line 109 – ‘into’ 4 groups. The change that the reviewer suggests has been made.

Line 112 – ‘were based on 15 minute time intervals’. The change that the reviewer suggests has been made.

Line 116-117 – As the data was freely available in the public domain, the requirement for informed consent was not necessary. The change that the reviewer suggests has been made.

Line 119 – amended. The change that the reviewer suggests has been made.

Line 122 – change to ‘pacing profile’. The change that the reviewer suggests has been made.

Line 124 – delete ‘percentage’ The change that the reviewer suggests has been made.

Line 125 – delete ‘percentage’ The change that the reviewer suggests has been made.

Line 130 – ‘the marathon for each individual athlete’ The change that the reviewer suggests has been made.

Results:

Line 146 – delete ‘per’. The change that the reviewer suggests has been made.

Line 148 – in men, no statistically significant differences were found… The change that the reviewer suggests has been made.

Line 150 – ‘Amongst women, we observed a large number of differences between groups which were statistically significant’ The change that the reviewer suggests has been made.

Line 153 – change men to male, and ‘the spits within which significant differences were found’.

Line 157 – change practically to almost. The change that the reviewer suggests has been made.

Line 161-162 – change implied to implies and ‘displayed a higher speed relative to mean overall race speed’. The change that the reviewer suggests has been made.

Table 2 – statistically significant but small effect size. Worthy of comment? Yes, we have decided to include a brief commentary about this (line 157-158). 

Line 165 – change confirmed to displayed. The change that the reviewer suggests has been made.

Line 168 – change for to ‘by’ , a to ‘the’, and beyond to ‘faster than’. The changes that the reviewer suggests have been made.

Line 169 – ‘a reduction in speed below average…’ The change that the reviewer suggests has been made.

Line 173 – change corroborates to suggests. The change that the reviewer suggests has been made.

Line 174 – change ‘of’ to ‘in’. The change that the reviewer suggests has been made.

Line 176 – delete second ‘per’. The change that the reviewer suggests has been made.

Line 177-8 – Tables 4 and 5 suggest that EP was the most commonly displayed pacing profile in both genders and at all levels of performance. This was followed by PP.’ 

Delete ‘regarding this’ The change that the reviewer suggests has been made.

Discussion:

Line 185 – we observed no differences in RP… The change that the reviewer suggests has been made.

Line 187 – change showed to demonstrated. The change that the reviewer suggests has been made.

Line 190 – delete ‘per’. The change that the reviewer suggests has been made.

Line 196 – change of to in. The change that the reviewer suggests has been made.

Line 198 – change with to from and which to where. The change that the reviewer suggests has been made.

Line 199 – ‘a previous study by…’. The change that the reviewer suggests has been made.

Line 200 – ‘with EP being the most common profile amongst the best runners’. The 

change that the reviewer suggests has been made.

Line 201 – ‘it is important to emphasise that a different course topography can influence observed pacing behaviours’ The change that the reviewer suggests has been made.

Line 204-205 – delete ‘which involved different competition profile’ The change that the reviewer suggests has been made.

Line 205 – the womens elite group was able to maintain a RP above average until … The change that the reviewer suggests has been made.

Line 209 – delete experimented and replace with ‘ experienced a RP drop earlier in the race’ The change that the reviewer suggests has been made.

Line 211 – change develop to display. The change that the reviewer suggests has been made.

Line 214 - showed ‘a similar…’ The change that the reviewer suggests has been made.

Line 217 - in ‘a’ marathon The change that the reviewer suggests has been made.

Line 218 – delete second ‘per’ The change that the reviewer suggests has been made.

Line 221 – the most common profile The change that the reviewer suggests has been made.

Line 225 – an increase in RPE. The change that the reviewer suggests has been made.

Line 228-9 – Diaz et al propose to aim for EP as an optimal strategy. The change that the reviewer suggests has been made.

Line 230-231 – reword this sentence please. This sentence has been remade (line 227-228). 

Line 233 – ‘that the majority of runners display EP..’ The change that the reviewer suggests has been made.

Line 237 – St Clair Gibson. The change that the reviewer suggests has been made.

Line 241 – change developed to displayed. The change that the reviewer suggests has been made.

Line 246 – delete ‘because’. The change that the reviewer suggests has been made.

Line 250 – you need to explain this statement. One explanation to this statement has been showed from line 248 to 250.

Line 253 – is there a word missing after ‘establish’? This error has been corrected. 

Line 258 – assessed is probably a better word than valued here. The change that the reviewer suggests has been made.

Line 270 – replace ‘on’ with ‘in’. The change that the reviewer suggests has been made.

Line 276 – If shorter splits had been used, we may have found greater variability. The change that the reviewer suggests has been made.

Line 277-8 – ‘we did not consider the effect of running in a group’ The change that the reviewer suggests has been made.

Line 281 - ‘variated’? I do not know this word The change that the reviewer suggests has been made.

Line 282 – change inversely to conversely and ‘may have benefitted from a greater density of runners…’ The change that the reviewer suggests has been made.

Line 291 – ‘well-trained’ and change develop to display. The change that the reviewer suggests has been made.

Line 292 – delete ‘of’. The change that the reviewer suggests has been made.

Line 293 – replace ‘points out’ with ‘indicates’. The change that the reviewer suggests has been made.

---

## [Editor Report · Decision Letter 2]

13 Jul 2020

Different race pacing strategies among runners covering the 2017 Berlin Marathon under 3 hours and 30 minutes

PONE-D-20-04748R2

Dear Dr. Mecías Calvo,

We’re pleased to inform you that your manuscript has been judged scientifically suitable for publication and will be formally accepted for publication once it meets all outstanding technical requirements.

Kind regards,

Daniel Boullosa

Academic Editor

PLOS ONE
---

## [Editor Report · Acceptance letter]

16 Jul 2020

PONE-D-20-04748R2 

Different race pacing strategies among runners covering the 2017 Berlin Marathon under 3 hours and 30 minutes 

Dear Dr. Mecías Calvo:

I'm pleased to inform you that your manuscript has been deemed suitable for publication in PLOS ONE. Congratulations! Your manuscript is now with our production department. 

Kind regards, 

on behalf of

Dr. Daniel Boullosa 

Academic Editor

PLOS ONE